



# Diversity II water quality parameters for 300 lakes worldwide from ENVISAT (2002-2012)

Daniel Odermatt[1,2], Olaf Danne[2], Petra Philipson[3], Carsten Brockmann[2]

[1]Odermatt & Brockmann GmbH, Zurich, 8005, Switzerland
[2]Brockmann Consult GmbH, Geesthacht, 20502, Germany
[3]Brockmann Geomatics Sweden AB, Kista, 164 40, Sweden

*Correspondence to*: Daniel Odermatt (daniel.odermatt@odermatt-brockmann.ch)

## 1. Abstract.

The use of ground sampled water quality information for global studies is limited due to practical and financial constraints.
Remote sensing is a valuable means to overcome such limitations and to provide synoptic views of ambient water quality at appropriate spatio-temporal scales. In past years several large data processing efforts were initiated to provide corresponding data sources. The Diversity II water quality dataset consists of several monthly, yearly and 9-year averaged water quality parameters for 340 lakes worldwide and is based on data from the full ENVISAT MERIS operation period (2002-2012). Existing retrieval methods and datasets were selected after an extensive algorithm intercomparison exercise using in situ
reference measurements for more than 40 lakes representing a wide range of bio-optical conditions. Chlorophyll-*a*, total suspended matter, turbidity, coloured dissolved organic matter, lake surface water temperature, cyanobacteria and floating vegetation maps, as well as several auxiliary data layers, provide a generically specified data basis that can be used for assessing a variety of locally relevant ecosystem properties and environmental problems. We demonstrate the use of the products by illustrating and discussing remotely sensed evidence of lake-specific processes and prominent regime shifts
documented in literature. The Diversity II data are available from https://doi.pangaea.de/10.1594/PANGAEA.871462, and Python scripts for their analysis and visualization are provided at https://github.com/odermatt/diversity/.

## 2. Introduction

Freshwater ecosystems have undergone more dramatic changes than any other type of ecosystems (Sectretariat of the Convention on Biological Diversity, 2010). Lakes contain about 87% of all surface freshwater (Gleick, 1996). The major
threats that affect lakes and reservoirs are water level changes, toxic pollution, salinization, eutrophication, acidification, sediment pollution and invasion of exotic species. Several upstream anthropogenic activities are related to these threats, such as agriculture, forestry, grazing, mining, irrigation, urbanisation and dams, hydraulic engineering and industrial development. All these pressures are interconnected and act concurrently to reduce water quality and contribute to the deterioration of the ecosystem, including habitat loss and reduced biodiversity.





The conditions of inland waters vary over a wide range of spatial and temporal scales, leading to important logistic and economic difficulties to monitor them on a regular basis. Some countries have national or regional lake monitoring programmes, which are primarily based on ground surveys. However ground surveys often fail to sample on appropriate spatial and temporal scales. Other countries do not have monitoring programmes due to a lack of funds. The use of satellite

remote sensing is a potentially cost-effective and efficient way to supplement the conventional in-situ point sampling surveys. Remotely sensed products for water availability and quality are complementary to in-situ data in terms of spatial and temporal coverage. They provide synoptic views of spatial distribution unachievable by other means, and are ideally suited to cover the broad range of space and time scales associated with inland water applications. However, remote sensing is limited in terms of parameter coverage and depth resolution.

The Medium Resolution Imaging Spectrometer (MERIS) was operated by the European Space Agency (ESA) in 2002-2012 and demonstrated unparalleled capabilities for water quality remote sensing. Extensive reviews of popular retrieval methods revealed a wide range of different algorithms, but the usage of MERIS data prevailed (Matthews, 2011; Odermatt et al., 2012). The first globally representative lake water quality dataset from remote sensing provided a snapshot of chlorophyll-*a* (CHL-*a*) concentrations in 80'000 lakes worldwide based on MERIS Full Resolution (FR) data acquired in 2011 (Sayers et

al., 2015). However, this dataset was compiled with an algorithm optimized for ocean colour remote sensing, whose suitability for inland waters was disavowed on several occasions (see e.g. Mobley et al., 2004; Morel and Prieur, 1977). Therefore we carried out an intercomparison of well-known and publicly available algorithms for the retrieval of CHL-*a* and other water quality parameters in optically complex waters with a heterogeneous reference dataset for more than 40 lakes (Odermatt et al., 2015a). The Diversity II water quality dataset is produced with the most suitable retrieval methods

identified through these investigations.

In addition to the optimized methodology, the Diversity II water quality dataset excels previous work by covering the full MERIS operation period with monthly, yearly and 9-year product aggregates, and a several additional water quality parameters. Hence it provides a generically specified data basis that can be used for assessing a variety of locally relevant ecosystem properties and environmental problems. Several case studies are available that demonstrate such assessments with

lake-specific foci (Odermatt et al., 2015b), but the larger part of the dataset is yet to be exploited.

### 3.  Input data

### 3.1. Geographical scope

We selected 340 lakes for processing (Figure 1) based on their biodiversity relevance, size, auxiliary and reference data availability, geographic distribution and particular user requests. 66 of those lakes are at least 50 km$^2$ large and located

within Ramsar Wetlands or listed as LakeNet Biodiversity Priority sites (www.worldlakes.org). The data table (https://doi.pangaea.de/10.1594/PANGAEA.871462?format=html#download) allows for their identification. The dataset includes 250 of the world's 350 largest lakes by extent, whereas size implies regional relevance and favours the feasibility of



remote sensing retrievals in general and of Lake Surface Water Temperature (LSWT) in particular (Politi et al., 2016). Various contributors provided in situ water quality measurements for 42 lakes, which are used as reference sites for quality assessment (Odermatt et al., 2015a). The largest reservoirs in South America and individual sites in Asia and Australia are included in order to improve the global representativeness. 50 additional lakes are included due to specific stakeholder requests.

In principle, the Diversity II water quality dataset could be extended to a much larger number of lakes. Size is the most important restriction in this regard, with a contiguous open water surface of roughly 1 km by 1 km being the theoretical minimum, but certain complications occurring even for larger water bodies. The total number of suitable lakes worldwide is expected to be between the 80'000 demonstrated by Sayers et al. (2015), and, neglecting shape properties, the 350'000 lakes larger than 1 km$^2$ identified by Verpoorter et al. (2014).

### 3.2. ENVISAT MERIS L1B FSG imagery

MERIS was operated in 2002-2012 on-board the near-polar orbiting ENVISAT satellite by the European Space Agency (ESA; Rast et al., 1999). It measured reflected solar radiance in 15 narrow spectral bands across visible and near-infrared (NIR) wavelengths. In FR mode, its push-broom charge-coupled device (CCD) arrays sampled the 1150 km wide swath at approximately 260 by 290 m ground resolution in across track and along track direction, respectively. MERIS had a nominal revisit time of 2-3 days at the equator and less at higher latitudes, but FR data was not systematically acquired in the early years until 2005, and in later years it varied slightly due to mission operations, and therefore the availability of usable data varies regionally and temporally (Figure 1).

We refer to three widely, but not consistently, used satellite image processing levels, in which Level 1 (L1) consists of Top-Of-Atmosphere (TOA) signals, Level 2 (L2) includes derived geophysical quantities and Level 3 (L3) represents spatio-temporally aggregated data. Approximately 300'000 MERIS L1 images were used as input for the production of the Diversity II water quality dataset. The data represents calibrated TOA radiance, also referred to as at-sensor radiance. It emerged from the 2014 bulk reprocessing using MERIS Instrument Processing Facility version 6. Its geo-orthorectification was improved using the Accurate MERIS Ortho-Rectified Geolocation Operational Software (AMORGOS; Bourg and Etanchaud, 2007), thus the data is referred to as MERIS L1B Full-Swath Geo-corrected (FSG).

### 3.3. AATSR ARC-Lake LSWT products

The Diversity II water quality dataset includes Lake Surface Water Temperature (LSWT) products that were readily provided by the ESA ARC-Lake project as version 3 production. The LSWT retrieval was performed with an optimal estimation approach (MacCallum and Merchant, 2013, 2012). Lake-specific prior surface temperatures were generated using an iterative scheme that is initiated with the monthly MODIS land and sea surface temperature climatologies. The ARC-Lake processor then uses valid satellite observations, simulations with the FLake model (Mironov, 2008) and Data Interpolating





Empirical Orthogonal Function (DINEOF; Alvera-Azcárate et al., 2005) techniques to iteratively create from this spatially and inter-annually invariant initial guess a field of spatially resolved temperature fields.

Several product types using different processing techniques, spatial and temporal aggregations are available (MacCallum and Merchant, 2014). We selected the DINEOF reconstructed, day- and night-time acquired monthly products in 0.05° spatial

resolution, whose file names are ALIDXXXX_PLREC9D_TS012SR.nc and ALIDXXXX_PLREC9N_TS012SR.nc, respectively, with XXXX being a four-digit lake ID. They are available for 298 out of the 340 lakes considered, and an empty LSWT product layers are contained in the remaining 42 lakes.

### 3.4. Auxiliary data

Each lake's perimeter was defined in a shapefile that resulted from vectorized outlines of the Synthetic Aperture Radar

Water Bodies (SAR-WB) map created by Santoro and Wegmüller (2014). These perimeters represent the maximum extent of water available from ENVISAT-ASAR acquisitions between 2002-2012, and each polygon's area and circumference were added to an attribute table. The polygons are intersected with the Global Lakes and Wetlands Database (GLWD; Lehner and Döll, 2004) Level 1 dataset, and ambiguities were manually resolved. The merged tabulated attributes are available in a metadata list (https://doi.pangaea.de/10.1594/PANGAEA.871462?format=html#download). Alternative lake names were

added to the list at every opportunity, but are neither exhaustive nor tracked.

Lake water surface level data (Crétaux et al., 2011) provided by the Laboratory of Studies on Spatial Geophysics and Oceanography (LEGOS) through their Hydroweb portal (http://www.legos.obs-mip.fr/soa/hydrologie/hydroweb/) was originally complementing and distributed with the Diversity II database. This data come as 1D discrete time samples, as opposed to the 2D temporal aggregated water quality maps. Furthermore, it is based on independent developments and

updates that would require continuous mirroring. Due to these differences, we refrained from adding them to the Pangaea Diversity II repository.

### 4. Data processing methods

The bulk production of temporally aggregated water quality parameters from L1B and auxiliary data requires a combination of several methods in an unsupervised processing chain. For this purpose we implemented the *CaLimnos* v1 processing chain

(Figure 2) for deployment on ESA's Earth observation data processing cluster *Calvalus* (Fomferra et al., 2012). It is composed of several processors for the ESA BEAM Toolbox (Fomferra and Brockmann, 2005), which has recently evolved into the Sentinel Application Platform (SNAP). The same input and auxiliary data, pre- and post-processing modules were also used to create 10-day aggregates for the investigation of phenological cycles in Lake Balaton (Palmer et al., 2015), and corresponding *CaLimnos* v1 L2 intermediate outputs were used for assessing the spatio-temporal variability of CHL-*a* in

Lake Geneva (Kiefer et al., 2015).



### 4.1. Pre-processing

The identification of pure water pixels is an essential pre-processing step, because even sub-pixel signal contributions from land surfaces can strongly affect the retrieval procedures, especially when using band arithmetic algorithms that do not check for input signal compliance at runtime. The Idepix algorithm is an open-source SNAP processor and performs such

identification for clouds, cloud shadows, cloud buffers, land, snow/ice, sun glint and mixed pixels (Danne, 2016) based on bottom-of-Rayleigh reflectance (BRR; Santer et al., 1999). BRR is subject to a partial correction of atmospheric effects, representing reflectance at the hypothetical boundary between an infinitesimally small aerosol layer and gaseous air layers above. It is the preferred signal when background reflectance for the estimation of aerosol optical thickness is highly uncertain, therefore BRR intermediate products are also used for the identification of shallow water areas and as input for the

Maximum Peak Height processor (Matthews et al., 2012) according to Figure 2.

Idepix uses the Shuttle Radar Topography Mission (SRTM) Water Body Dataset (SWBD; Slater et al., 2006) as a static a priori land-water mask, which is a snapshot of global water surface extent between 56°S and 60°N in February 2000. It applies several arithmetic expressions, a spectral unmixing algorithm for mixed pixel identification, and two back-propagation Neural Networks (NN) for cloud identification to MERIS FSG L1B and BRR input data (Kirches et al., 2013).

Output is a pixel identification flag layer which is much better suited for water constituent retrieval than the original L1B product flags (Ruescas et al., 2014). However, usage with inland waters is subject to two particular challenges. First, Idepix' sea ice identification uses climatological auxiliary data that is not available for lakes, therefore lake ice identification is less accurate. Second, ephemeral water surfaces that may extend far beyond the SRTM observed extent are always clipped to the latter.

Bottom visibility is a critical and unmastered error source for water quality retrieval, because most algorithms that provide concentrations of water constituents do not account for benthic reflectance contributions in these so-called optically shallow waters. In fact, a pre-conditions for them is optically deep water (i.e. no bottom reflection). Sandy or vegetated substrates cause surface-leaving signals that can closely resemble increased suspended sediment and phytoplankton concentrations in the water column, respectively, and thus distort retrievals. Only very few algorithms are actually dealing with the detection

of optically deep water, and none of them applies to inland waters. Based on recommendations for clear coastal waters (Cannizzaro and Carder, 2006) and own investigations, we defined a band ratio that evaluates the relative elevation of oligotrophic lakes' 555 nm water-leaving reflectance peak, but using BRR in three MERIS bands as input due to the lack of robust automated atmospheric correction algorithms for such conditions (Eq. 1; Odermatt et al., 2015a).

$$IF\ ratio\_490 = \frac{BRR_{band3} \cdot BRR_{band7}}{(BRR_{band5})^2} < Thres, shallow = TRUE \qquad \text{Eq. 1}$$

Due to the ambiguity of certain substrates' shallow water reflectance and deep-water reflectances, this optical signature alone is prone to false positive identifications. It becomes much more robust when applied to temporally aggregated *ratio_490* due to the relative persistence of benthic features as opposed to the dynamically changing water composition. Summer half-year



mean averages were selected after evaluation of several statistical aggregation methods. Corresponding aggregates are composed of all cloud-free MERIS observations in May to October 2008 for the Northern hemisphere, and November 2008 to April 2009 for the Southern hemisphere. Lakes with constantly high turbidity, such as Lake Balaton, still trigger false positives. Therefore each *ratio_490* aggregate map was verified with high-resolution satellite imagery and bathymetry

information. Considerable shallow water areas in about 30 oligo- to mesotrophic lakes were masked using a threshold of 0.65, which, in the case of the Beaver Island Archipelago in Lake Michigan, masks areas that are between 5-10 m deep (Figure 3). Pixels removed in such manner are indicated in a separate product layer (*shallow,* Figure 3 bottom). In the Lake Michigan example, there are some patterns in the *chl_fub* product that still resemble bathymetry features, but their concentration levels are within variations for deep water areas apart from a few individual pixels. Especially in more turbid

lakes, lower thresholds are applied to prevent false positives according to the corresponding column in the lakes list (https://doi.pangaea.de/10.1594/PANGAEA.871462?format=html#download).

## 4.2. Water quality retrieval

CHL-*a* retrieval in optically complex waters is straightforward when using the secondary reflectance peak at red and near-infrared (NIR) wavelengths (e.g. Gitelson, 1992; Gons, 1999; Gower et al., 1999). However, using MERIS observations this

peak is only accessible in moderately productive or turbid waters, while clear and humic waters call for different approaches (Odermatt et al., 2012). Moore et al. (2014) developed an Optical Water Type classification (OWT) framework, which supports the distinction of these different water types, and which is available as SNAP plugin (Peters, 2016). It assigns water-leaving reflectance spectra to seven end members, which were identified through cluster analysis of in situ measurements. Classes 1-3 represent clear or absorbing waters, classes 4-5 represent high phytoplankton and classes 6-7

represent high suspended mineral contents (Figure 4). The OWT algorithm depends on the accurate correction of atmospheric effects (Eleveld et al., 2017), which was assessed by classifying 42 matchup pairs of in situ reflectance measurements in 10 diverse lakes and MERIS water-leaving reflectance from various atmospheric corrections. Water-leaving reflectance obtained with the CoastColour NN algorithm (description below) achieved the best agreement, in which half the matchup pairs were assigned to the same OWT, and adjacent classes were assigned in 14 cases (Odermatt et al.,

2015a). In 5 out of the remaining 7 cases, the optically quite similar classes 1 and 3 are confused. This mismatch due to differences between in situ measured and satellite observed reflectance is quite significant. However, when considering only the separation between classes 1-3 and 4-7, and thus the feasibility of CHL-*a* retrieval based on the secondary reflectance peak, the approach becomes very robust, with only 2 out of the 42 pairs being confused. The OWT maps therefore provide a rough but robust indicator for CHL-*a* algorithm selection. Most lakes are relatively clearly dominated by either OWT 1-3 or

4-7, which makes the CHL-*a* product selection straightforward. In rare cases like Lake Turkana (Figure 5), such a selection can only be made if either the lower or the upper end of the dynamic range is considered more relevant. Otherwise, it is recommended to either split the lake perimeter or merge the CHL-*a* products e.g. by weighting them with turbidity levels.





For the Diversity II production the Maximum Peak Height algorithm (MPH; Matthews et al., 2012) was developed further and implemented in a SNAP operator (Block, 2016) because it outperformed other red-NIR reflectance peak algorithm in the algorithm intercomparison study (Matthews and Odermatt, 2015; Odermatt et al., 2015a). It uses BRR in MERIS bands 6-10 and 14 for the retrieval of the red-NIR reflectance peak height and position, which allow for the identification of

cyanobacteria and eukaryote dominated pixels, water surface covering by cyanobacteria scum or floating vegetation, and CHL-*a* quantification. Dedicated empirically calibrated equations are used for the retrieval of CHL-*a* concentrations in eukaryote and cyanobacteria dominated waters. Technically, the algorithm is designed to cover the range of 0-1000 mg/m$^3$ CHL-*a*. However, retrieval accuracy is significantly better for lakes that are predominantly OWT 4-7, namely eutrophic to hypertrophic waters.

The FUB algorithm (Schroeder et al., 2007), named after the Free University of Berlin, is a bundle of dedicated NN algorithms for CHL-*a*, Total Suspended Matter (TSM) and Coloured Dissolved Organic Matter (CDOM) retrieval from MERIS L1B data, and a fourth NN that computes AOT at four wavelengths (440, 550, 670, 870 nm) and water-leaving reflectance in all bands up to 708 nm, except at 680 nm. The algorithms are trained with radiative transfer simulations using the Matrix Operator Model (MOMO; Fell and Fischer, 2001) covering CHL-*a*, TSM and CDOM concentration ranges of

0.05-50 mg/m$^3$, 0.05-50 g/m$^3$ and 0.005-1 m$^{-1}$, respectively, and using  MERIS bands 1-7, 9, 10 and 12-14 as input. The training ranges are a severe limitation for global usage, but specific retrieval quality flags indicate for each of the four NN algorithms if the input or output exceeds the training range. However for oligo- to eutrophic and in particular humic lakes, which are commonly identified as OWT 1-3, the FUB algorithm's CHL-*a* output outperformed all other candidates in the intercomparison study (Odermatt et al., 2015a). Note that FUB uses shorter wavelengths that reach deeper into the water

column than MPH, which means that the two CHL-*a* products represent different depths and may not converge at intermediate concentrations (ca. 10-30 mg/m$^3$), where both algorithms produce valid results.

For the retrieval of TSM via particulate backscattering at 443 nm (bb_spm_443 in Figure 2) and turbidity, as well as water-leaving reflectance input for the OWT classification, we used the CoastColour NN algorithm. Its architecture is based on the approach described in Doerffer & Schiller (2007), with two dedicated NN systems performing atmospheric correction  and

inherent optical properties retrieval (Doerffer, 2011; Ruescas et al., 2014). In contrast to earlier NN algorithms, the CoastColour NN was trained with significantly larger concentration ranges, namely 0.03-1000 g/m$^3$ TSM and 0.03-500 mg/m$^3$ CHL. It was extensively validated with the CoastColour Round Robin data set (Nechad et al., 2015; available in Pangaea) and lake in situ measurements (Odermatt et al., 2015a).

### 4.3. Post-processing and auxiliary data

The aggregation of L2 to L3 products (Figure 2) facilitates temporal binning and collocation in a common coordinate grid with WGS 84 (EPSG 7030) coordinate system. Monthly aggregates are created using the input, output and aggregation methods listed in Table 1, and the same aggregation methods are used to create yearly and 9-year aggregates from monthly and yearly aggregates, respectively, which ensures that all months input aggregate periods are weighted equally even if the





numbers of L2 available in these periods may differ strongly. Aggregation of biophysical parameters is done using the mean of all valid nearest-neighbour input pixels for each output pixel, while the OWT L3 output layer consists of the most frequently observed class value across all available input layers, using the lower class in the rare case of a draw.

Monthly, yearly and 9-year aggregates for each lake are saved in individual GeoTIFF files, and compressed in 13 ZIP files representing 11 annual archives for the monthly aggregates, one archive for the yearly aggregate, and the 9-year aggregate. These 13 ZIP archives are zipped again to make each lake available for download in a single file of up to 18.9 Gigabyte size (Caspian Sea).

For extracting product statistics and for visualization of the products, a Python package is available at https://github.com/odermatt/diversity. The scripts included in the package allow for creating spatial and temporal plots such as shown in Section 5 (Figure 5 to Figure 12). They also feature the use of blacklists, e.g. to exclude all products with scarce lake extent coverage from further analyses.

**Table 1: Input, output and aggregation specifications for the monthly products.**

| Algorithm | L2 input layer(s) | Aggr. | L3 output layer(s) |
|---|---|---|---|
| MPH | chl: float [mg/m$^3$] | mean | chl_mph: float [mg/m$^3$] |
| FUB | algal-2: float [mg/m$^3$] | mean | chl_fub: float [mg/m$^3$] |
| FUB | yellow_subs: float [m$^{-1}$] | mean | cdom_fub: float [m$^{-1}$] |
| CoastColour | bb_spm_443 | mean | tsm_cc: float [g/m$^3$] |
| CoastColour | turbidity: floating numbers [FTU] | mean | turbidity_cc: float [FTU] |
| MPH | if CYANO_FLAG not FLOAT_FLAG: binary | mean | immersed_cyanobacteria: float [0-1, dl] |
| MPH | if CYANO_FLAG and FLOAT_FLAG: binary | mean | floating_cyanobacteria: float [0-1, dl] |
| MPH | if FLOAT_FLAG not CYANO_FLAG: binary | mean | floating_vegetation: float [0-1, dl] |
| OWT | dominant_class: integer [1-7, dl] | mode | owt_cc_dominant_class: integer [1-7, dl] |
| ratio_490 | See Section 4.1 | mean | shallow: binary |
| lswt_d | See Section 3.3 | none | lswt_d_mean: float [deg. K] |
| lswt_n | See Section 3.3 | none | lswt_d_mean: float [deg. K] |

## 5. Results

The Diversity II water quality datasets were used for several lake-specific assessments, most prominently for indicating fish assemblages and status assessments in Lake Vänern (Sweden; Sandström et al., 2016). Other use cases are described as *biodiversity stories* and made available from www.diversity2.info. A summary of three selected examples verifies how the remotely sensed parameters respond to spatio-temporally evident or relatively well-documented biophysical events.





### 5.1. Lake Biwa

Lake Biwa is a monomictic lake northeast of Kyoto (Japan), up to 104 m deep and with a surface area of 658 km$^2$ among the smaller lakes in the dataset. The primary productivity of the lake is relatively low, but subject to strong spatial gradients that are related to the distribution of residential and industrial areas, which are concentrated on the south-eastern shore and responsible for increased riverine Nitrogen input (Ohte et al., 2010) that increase near-shore phytoplankton growth (Figure 6). In December 2007 investigations with an AUV (Autonomous Underwater Vehicle) revealed more than 2000 dead organisms on the lake's bottom, mostly endemic Isaza gobi fish and lake prawns. Low dissolved oxygen concentrations of less than 1.0 mg/l near the lake bottom in November were identified as the main cause for an increased exposure of aquatic organisms to heavy metals and the die-off (Itai et al., 2012; Kawanabe et al., 2012). Oxygen supply depends on wintertime vertical mixing, which, aside from wind stress, depends on the vertical density gradients and thus thermal stratification. *In situ* temperature profiles from the Lake Biwa Environmental Research Institute's regular limnological survey program suggest that vertical mixing remained very weak in the winter of 2006/2007 (Kawanabe et al., 2012). Even though relating surface to bottom temperatures is not without caveats, significantly higher LSWT with spatially averaged 8.2°C is observed in March 2007 than in the other years (Figure 7), suggesting that minimal annual LSWT could be a valuable proxy for vertical mixing in temperate lakes.

### 5.2. Lake Nicaragua

Lake Nicaragua/Cocibolca is the largest lake in Central America with a surface area of 7851 km$^2$. It is polymictic with a maximum and average depth of only 26 and 15 m, respectively. It is subject to prevalent ecological issues such as untreated urban wastewater discharge and immissions from agriculture (soil erosion, fertilizer and pesticide immissions) and aquacultures that introduce non-resident Tilapia species and possibly novel diseases. Moreover, the planned construction of the Nicaragua Canal connecting the Caribbean Sea to the Pacific Ocean would bring about a significant shift in the lakes ecological status, most directly through the excavation of a 27.6 m deep, 520 m wide and 286 km long waterway across the centre of the shallow lake, which will strongly affect light availability within the water (Meyer and Huete-Pérez, 2014). In spite of the limited availability of MERIS FR data over Latin America (Figure 1), it can contribute to estimating baseline conditions prior to the intervention. Generally maximum and minimum turbidity occur around August and February, respectively (Figure 8, top), within a range between 2-20 FNU. Outliers such as in October 2005 and May 2007 can occur when only a small area of the lake is sampled. However, the 2011 turbidity peak in October is related to an extraordinary shift from cyanobacteria to eucaryotic algae (Figure 9), which comes with significantly lower CHL concentrations throughout the year from both the MPH (Figure 8, top) and the FUB algorithm (not shown), but also a second productivity peak. Even though data continuity and *in situ* measurements are required for further interpretation, the available data suggests that the lake was in a relatively unstable state at the end of the observation period.



### 5.3. Lake Victoria

Lake Victoria is the second largest fresh water lake in the world and is situated in a shallow depression between the Great Rift Valley and the western Albertine Rift, with a shoreline shared by Kenya, Uganda and Tanzania. It is up to 83 m deep, eutrophic and light-limited (Hecky et al., 2010), and it's thermocline is usually at around 30-40 m with complete mixing occurring once a year (MacIntyre et al., 2014; Payne, 1986). About 80% of the water input to Lake Victoria are from direct rainfall (Swenson and Wahr, 2009), and atmospheric deposition is the most important Phosphorous source in pelagic areas of the main basin (Tamatamah et al., 2005). In contrast, the Nyanza Gulf (also known as Winam or Kavirondo Gulf), the lake's most distinctive morphological feature in the north-east, receives about 10% of the lake's terrestrial inflow, and it was concluded from ground measurements between March 2005 and March 2006 that the Nyanza Gulf even received Phosphorous input from the main basin, in contrast to the paradigm that the gulf is a major contributor to the lake's increasing nutrient enrichment (Gikuma-Njuru et al., 2013). As a matter of fact, MERIS observations confirm that the most productive areas are located in the very east of the Gulf throughout 2005 (Figure 11) and for the first half of 2006 (not shown). During this period, the CHL levels in the lake's centre in July 2005 appear extraordinarily high. This observation can be verified through a comparison with the number of available observations (2-7) and the abundance of immersed cyanobacteria (0.4-1) in this area and month. This means that most parts of this cyanobacteria bloom were identified in several observations.

The Nyanza Gulf was also subject to intensive growth of water hyacinths (*Eichhorna crassipes*) in response to *El Niño* precipitation anomalies in 1998 (Albright et al., 2004) and 2007 (Fusilli et al., 2013). Figure 12 displays the 2007 proliferation event according to Diversity II data. The hyacinths appear in *floating_cyanobacteria_mean* rather than the expected *floating_vegetation_mean*, assumingly due to persistent cyanobacteria dominance in the Gulf. Given that Idepix is likely to mask completely overgrown pixels as land (see Section 4.1), the remaining water pixels counted for the abundance consist partly of immersed cyanobacteria, partly of floating eucaryotes. Despite these limitations and the fact that monthly aggregates lack the spatial details of individual observations, the extent and course of the proliferation matches well with the MODIS observations presented by Fusilli et al. (2013).

### 6.  Discussion

#### 6.1. Conclusions

The Diversity II dataset is the first globally representative, temporally resolved and methodologically consistent information source for inland water quality dynamics from satellite Earth observations. It includes monthly, yearly and 9-year temporally aggregated geophysical maps of various water quality parameters, which provide unprecedented possibilities for exploitation at global and local scale. Global analyses are yet to be carried out, with caution to the limitations mentioned hereafter. At local scale, several case studies demonstrated how the data could effectively contribute to traditional investigations of lake



specific processes and events. The Diversity II product user handbook (Odermatt et al., 2015b) helps to improve such interpretation of remotely sensed data by providing background knowledge of acquisition and retrieval methods, and Python scripts are available that facilitate standard information extraction and visualization from the individual GeoTIFF files.

The methods used for producing the Diversity II dataset represent the state-of-the art at the end of the ENVISAT era. It is a major asset of Earth observation that productions from L1 observations can be repeated as improved methods become available, so there is no doubt that the methods in use for Diversity II will be improved and overhauled in the future. Several lessons were learned for such repeated productions, some of them involving challenges for future research. For example, the binary identification of optically shallow waters and floating lake ice have received relatively little attention in recent years, but under certain circumstances they have much larger effects on the final product accuracy than retrieval algorithms. Our method for the identification of clouds, land and mixed pixels is more advanced, but it remains critical for all partly cloudy situations. Therefore, the development of such methods should receive much more attention, relative to the number of retrieval algorithms that were developed in recent years. The latest generation of retrieval algorithms will be based on further advanced water types (Spyrakos et al., n.d.) and ensemble approaches that account for the selection of multiple algorithms' estimates (as e.g. left to the user with *chl_mph_mean*/*chl_fub_mean*). Finally, new approaches are needed for the consolidation of such increasingly large geospatial datasets, and for the extraction of relevant information, which is still mostly based on lake-specific knowledge.

### 6.2. Limitations

Areas of melting lake ice are optically very similar to water, and the false retention of a lake ice covered pixel by the Idepix algorithm can lead to highly irregular constituent estimates. Indicators for such cases are the seasonal timing, sharp linear features in water constituent products, indicating lake ice borders or cracks, and very low values in mean LSWT. Auxiliary data from the NOAA/NSDIC Global Lake and River Ice Phenology Database could also help with the identification. A procedure to fix monthly products that are affected by melting lake ice artefacts using the BEAM L3 binner is described in Odermatt et al. (2015b).

Ephemeral (also intermittent or seasonal) lakes as well as other lakes and reservoirs that significantly change their extent over time are usually not well represented because the lakes' areas are clipped to the extent of the SWBD in February 2000. Their change in extent also complicates the identification of shallow water areas in a way that makes our approach based on temporally consistent spectroradiometric properties inapplicable. Therefore *ratio_490* thresholds for such lakes were set to 0.0 to disable shallow water identification. Furthermore, many of these lakes are subject to very high salinity levels and other typical habitat properties that favour extraordinary types of water constituents and accordingly bio-optical properties, which can significantly affect the validity of the water quality products. Particular care is thus needed when a lakes' product layer extent is significantly smaller than expected.



The relative abundances in the *floating_vegetation* and *floating_cyanobacteria* layers include only pixels that passed the foregoing Idepix masking. This means that especially very densely covered water pixels are previously identified as land pixels and not counted, resulting in an overall underestimation of abundances.

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



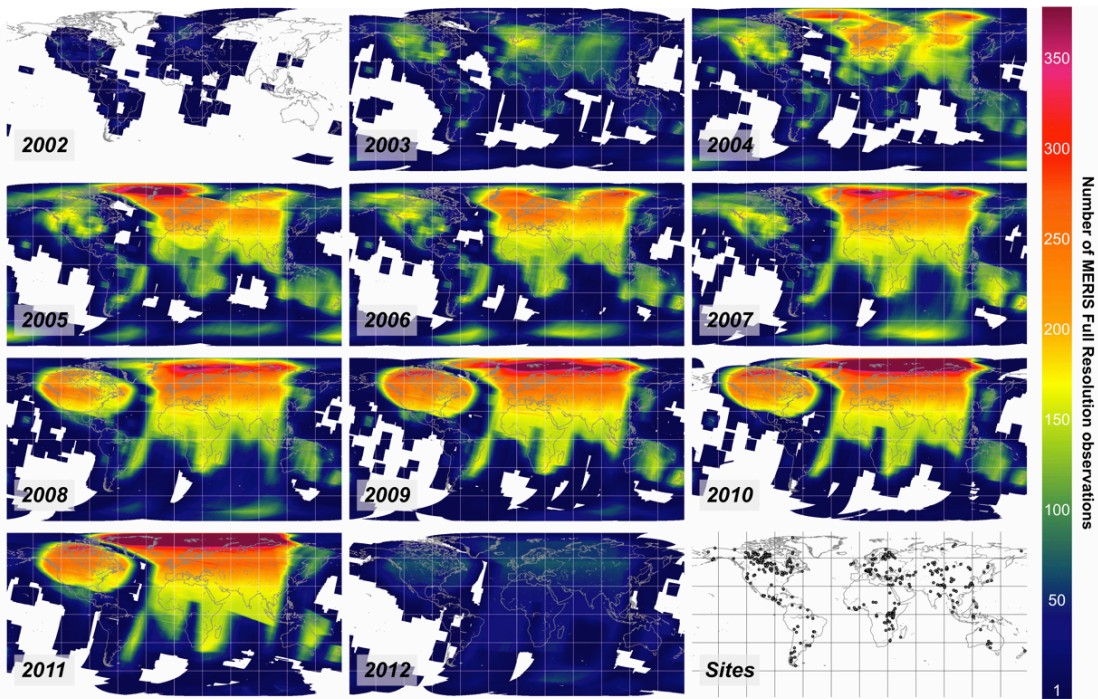

**Figure 1: Global density maps for the bulk reprocessed MERIS FR dataset in the years 2002-2012, and distribution of the 340 lakes available in the Diversity II water quality dataset (bottom right).**





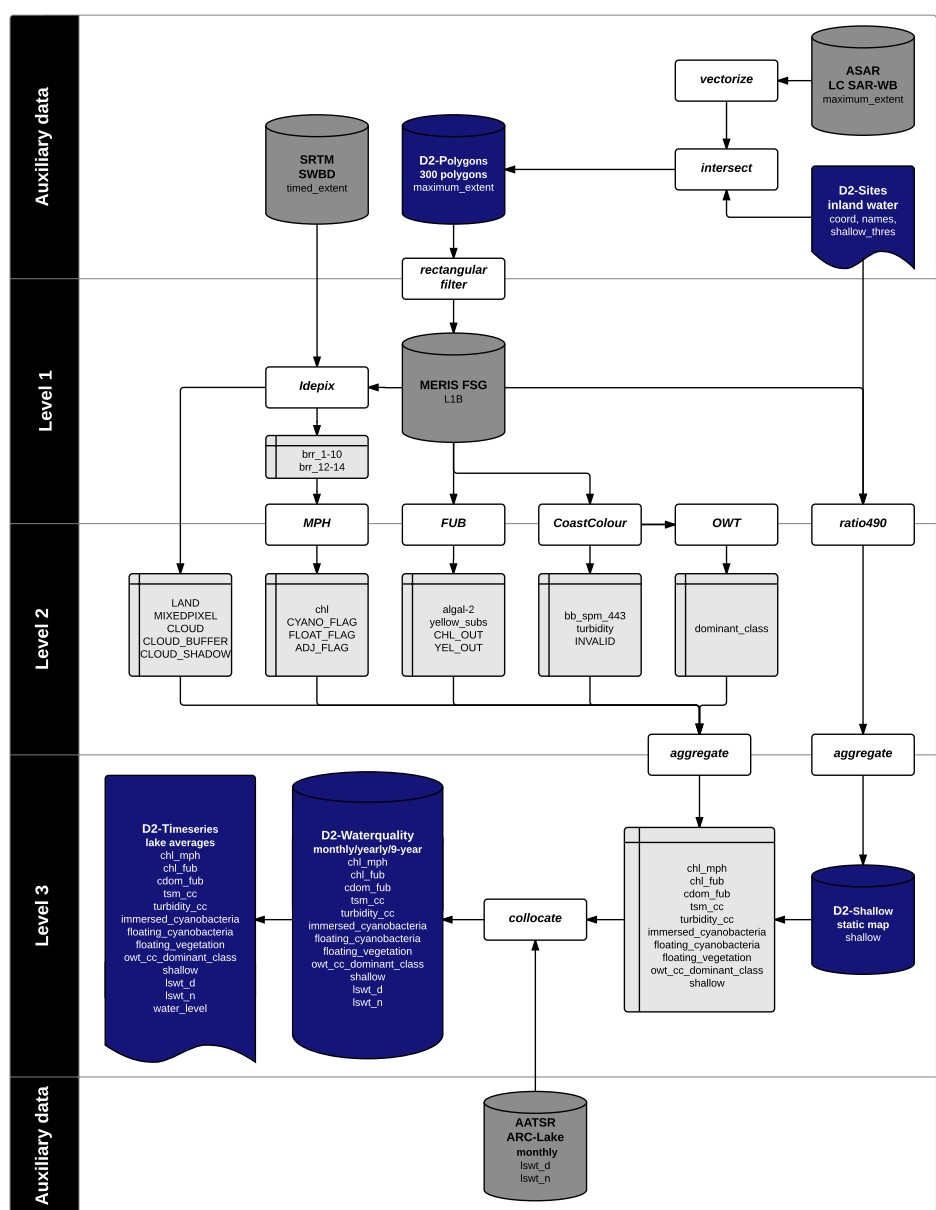

**Figure 2: The *CaLimnos* v1 processing chain for inland waters. Coloration indicates algorithms and downstream processes (white), input and auxiliary data (dark grey), intermediate products (light grey) and output products (blue).**



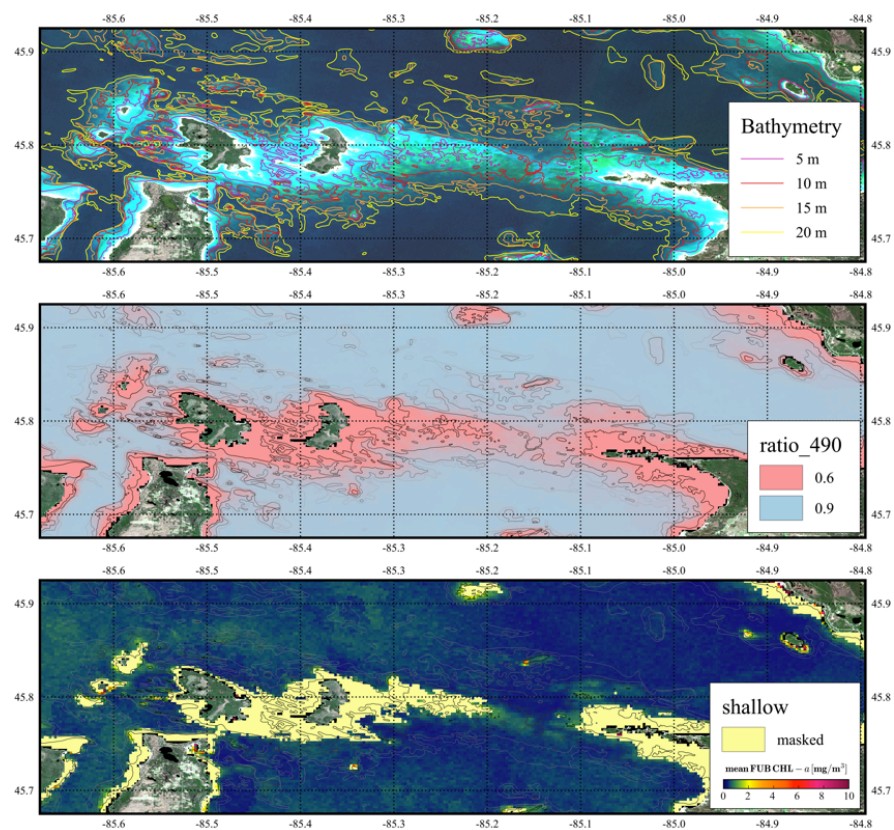

**Figure 3: Shallow water flagging for the Beaver Island Archipelago in the North of Lake Michigan. Top: Sentinel-2A true colour image, 8 May 2017. Centre: ratio_490 from MERIS data, acquired in May-October 2008. Bottom: Shallow water mask for ratio_490 with a threshold of 0.65 on top of the CHL layer for October 2011 as contained in the product layer *shallow*. Bathymetry data provided by NOAA-NCEI.**





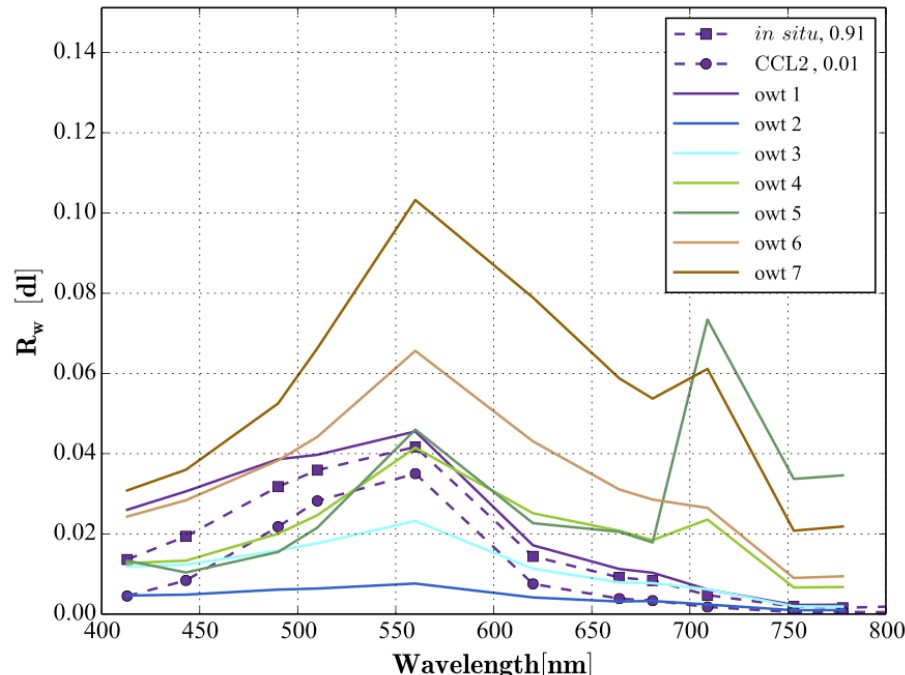

**Figure 4: OWT end member water-leaving reflectance spectra and a OWT 2 retrieval example for an in situ and MERIS CCL2 reflectance pair from Lake Zurich, 15 August 2007. Respective class membership scores are indicated in the legend.**

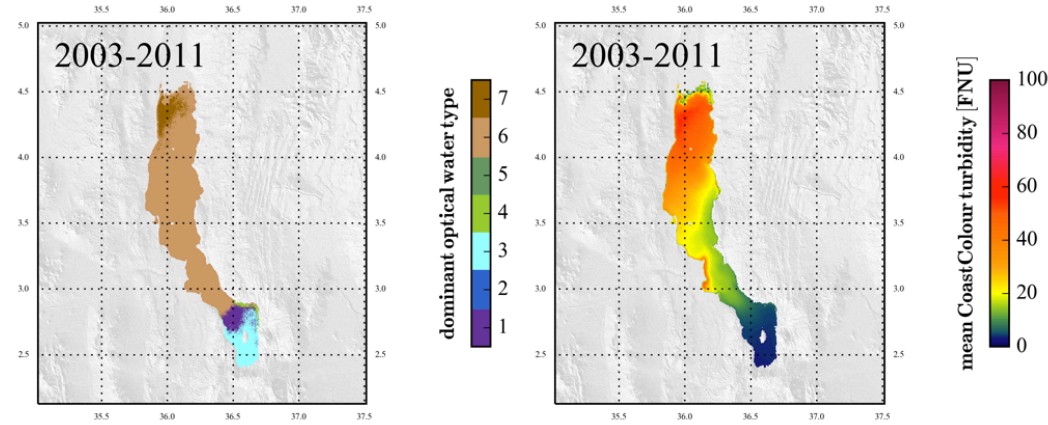



**Figure 5: 9-year aggregated OWT in Lake Turkana (left; *owt_cc_dominant_class_mode*), which features a very prominent gradient in turbidity (right; *turbidity_cc_mean*). Maximum turbidity and predominantly OWT 7 are observed in the North, where its main tributary, the Omo River, provides about 90% of the lake's inflow (Beadle, 1981). In contrast, the terminal basin in the South corresponds to OWT 1 and 3, which are 2nd and 3rd lowest in turbidity according to the end members in Figure 4.**

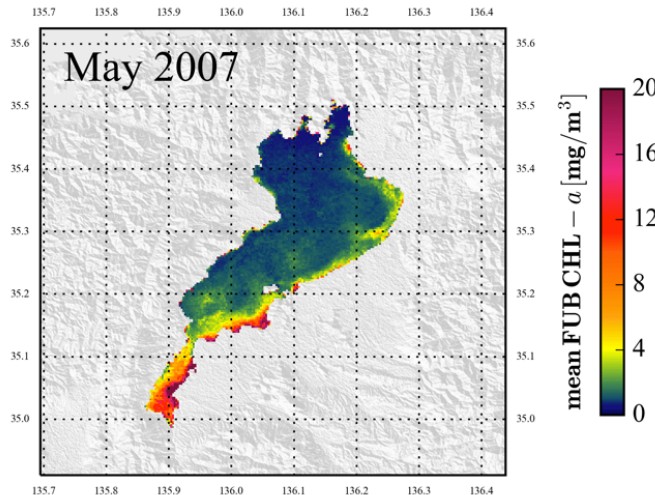

**Figure 6: CHL in Lake Biwa, May 2007, L3 aggregate of four cloud-free and one partly cloudy image (*chl_fub_mean*).**



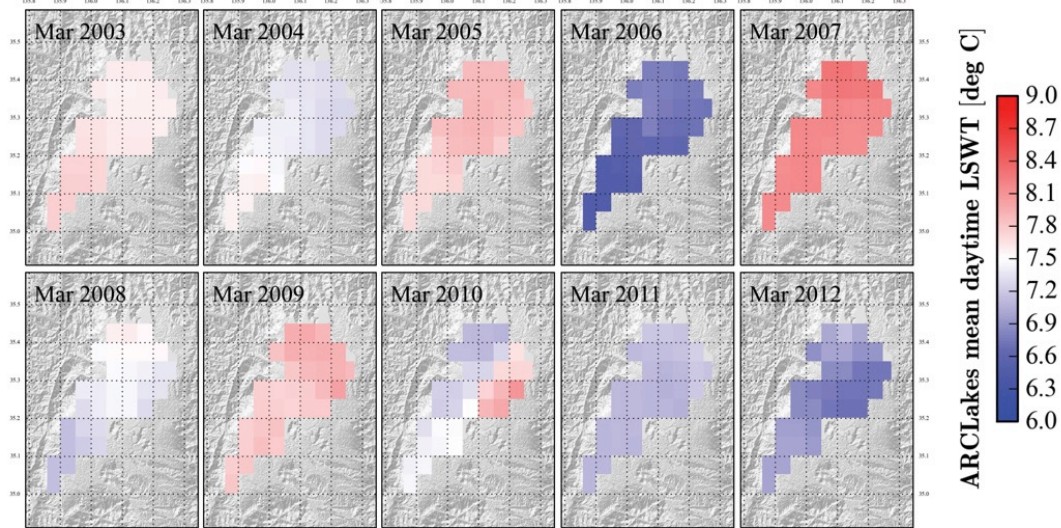

**Figure 7: LSWT in March in Lake Biwa (*lswt_n_mean*). The March LSWT mark the annual minimum for every year contained in the dataset.**



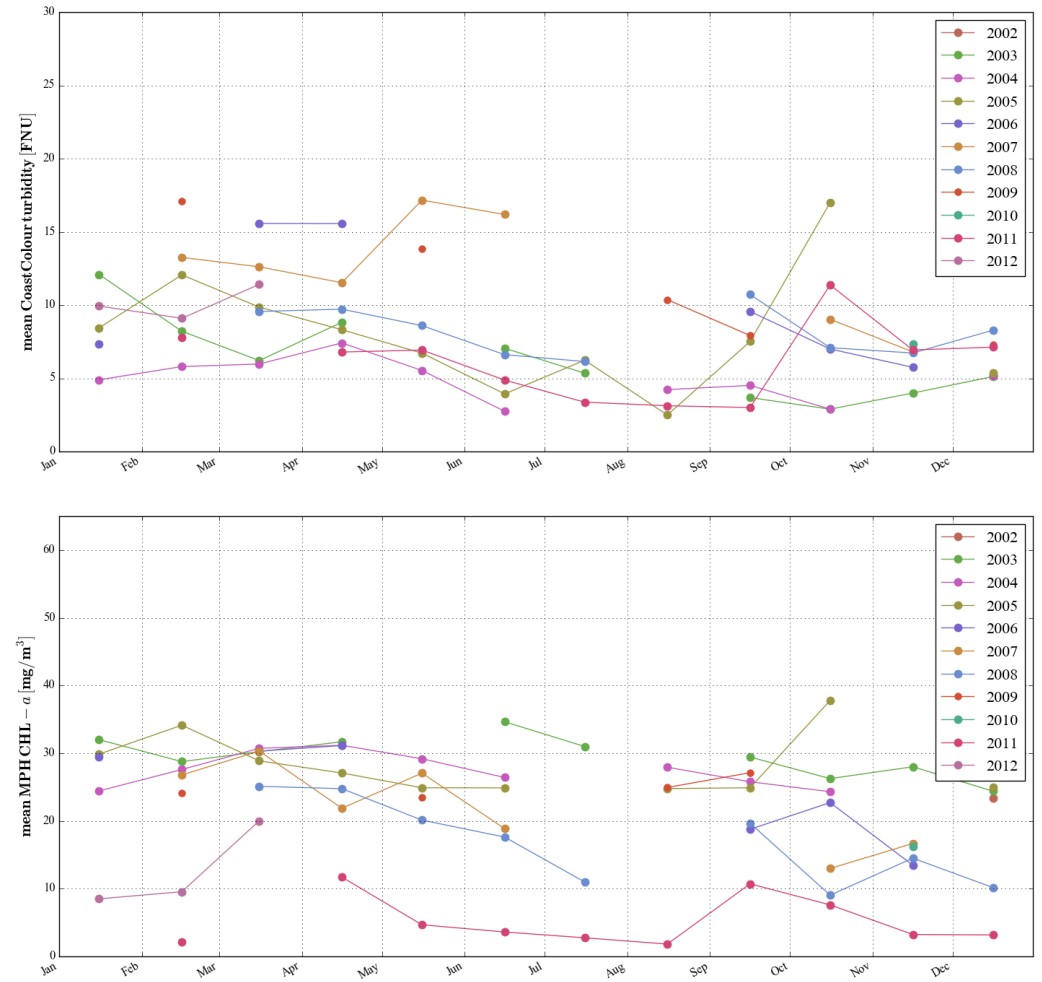

**Figure 8: Timeseries of spatially averaged Turbidity (top) and CHL (bottom) in Lake Nicaragua, with data gaps especially during the rainy season (June-October).**



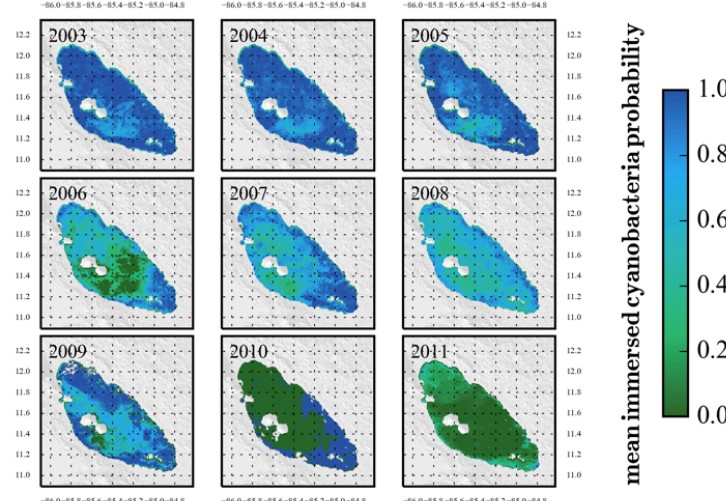

**Figure 9: Mean annual cyanobacteria probability in Lake Nicaragua, 2003-2011 (*immersed_cyanobacteria_mean*). Note that the number and distribution of valid observations across the years is quite unequal (Figure 10).**

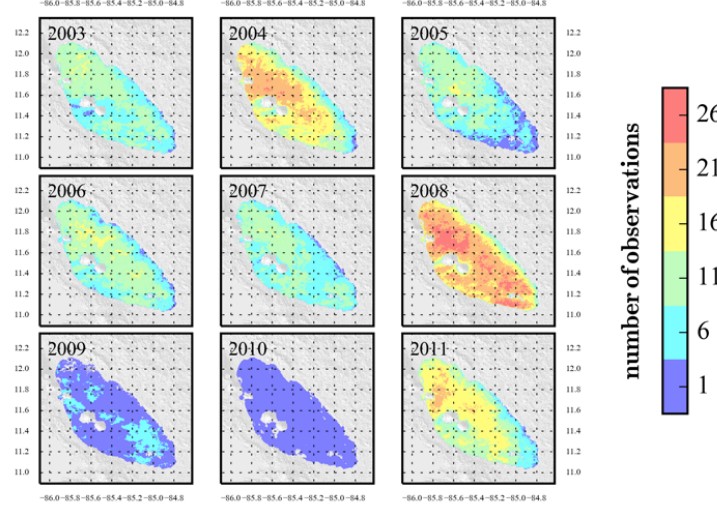

**Figure 10: Number of observations for the annual cyanobacteria abundance maps in Figure 9 (*num_obs*).**





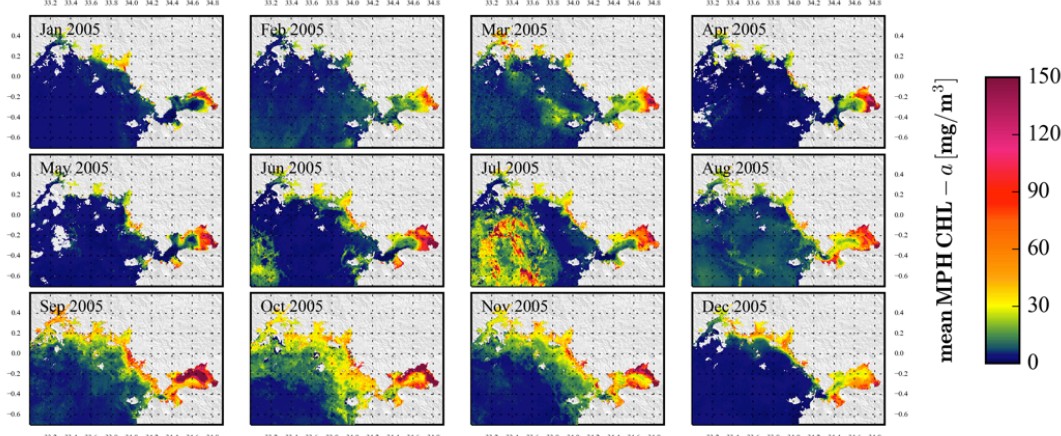

**Figure 11: Monthly CHL concentrations in north-eastern Lake Victoria, 2005 (*chl_mph_mean*).**





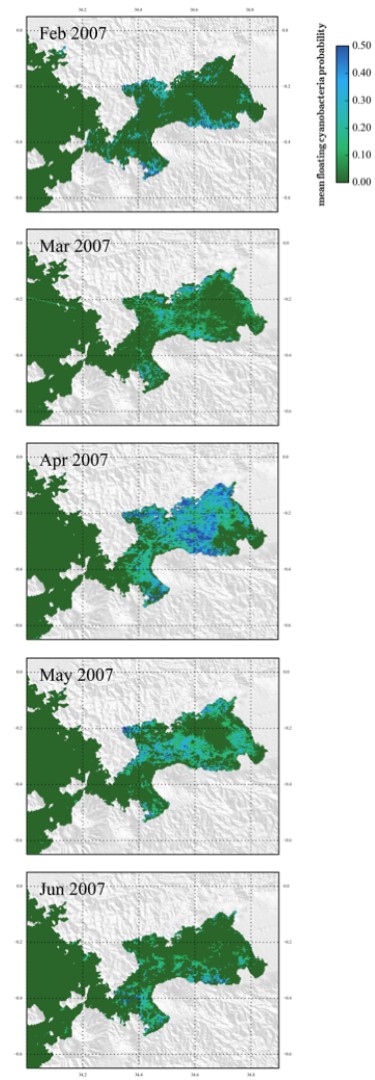

**Figure 12: Peak of the 2007 water hyacinth growth in Nyanza Gulf, Lake Victoria (*floating_cyanobacteria_mean*).**