# Peer review of "Diversity II water quality parameters for 300 lakes worldwide from ENVISAT (2002-2012)"

_Earth System Science Data, 2018_

## Referee Comment (RC1) · Anonymous Referee #1 · 30 Jan 2018

General assessment This manuscript describes several ways/approaches to obtaining water parameters for the Diversity II water quality dataset. The authors used a large amount of satellite images to generate the dataset. It is of high value for application. Definitely, we can produce geophysical data from satellites. Of high concern is how the data accuracy is. It is very necessary to document the data accuracy with at least one index. Without the document, users cannot grantee the reliability of his/her analyzed results. Technically, the algorithms used are threshold dependent that often artificially determined. The threshold may vary with images (Liu et al. 2012). It is logical to briefly clarify the data accuracy with the threshold issue for readers to use the dataset.

Specific comments Page 2 Line 22: please delete the 'a' ahead of 'several'. Page 3 Line 14: please provide the full spelling of 'FR'. Page 3 line 29: change the order between

2013 and 2012. Page 4 line 18: complemented? Page 5 line 22: pre-condition/s? Page 7 line 31: add 'the' ahead of 'WGS'.

[Liu et al. 2012. A physical explanation of the variation in threshold for delineating terrestrial water surfaces from multi-temporal images: effects of radiometric correction. International Journal of Remote Sensing, 33, 5862-5875]

---

## Referee Comment (RC2) · Anonymous Referee #2 · 21 Feb 2018

General comments This manuscript reports a valuable dataset (Diversity II water quality dataset) that was mainly generated from ENVISAT/MERIS data for evaluating water quality in more than 300 lakes worldwide. It can be considered that the dataset will be very useful to a wide range of users, especially in some lakes where in situ water quality data are unavailable. However, since the present manuscript lacked accuracy and available data frequency assessments for the dataset, it is hard to win users' confidence in their applications. Therefore, it is suggested that the authors can provide this kinds of information in their manuscript.

Specific comments Page 1, line 12. "The Diversity II water quality dataset consists of several monthly, yearly and . . .". It will become better if the authors can provide a summary for availability (or temporal frequency) of each water quality parameter.

[Figure]

For example, in how many lakes and in a given year users can obtain water quality parameters in every month, or in every two month, and so on. This information can help users to judge the dataset is suitable for their applications or not.

Page 2, lines 19-20, the authors wrote that "The Diversity II water quality dataset is produced with the most suitable retrieval methods identified through these investigations." Nevertheless, it is better to provide several indices such as RMSE, NMAE (normalized mean absolute error), and so on for assessing accuracy of the dataset.

Page 8, Table 1. It is suggested that the authors include atmospherically corrected remote-sensing-reflectance in their dataset because it can make users flexibly choose different retrieval algorithm (e.g., OC4E, two- or three-band models) to estimate water quality parameters according to characteristics of their lake.

Technical corrections Page 9, line 25. "Generally maximum and minimum turbidity occur around August and February, respectively..." should be "February and August, respectively...", right?

Page 9, line 29. "...from both the MPH (Figure 8, top)..." should be "(Figure 8, bottom)", right?

Page 10, section 6. It is better to separate discussion and conclusions into different section. In addition, the section of conclusions should be the last one. However, if this is the style of the journal, please ignore this comment.

Page 11, line 25. Please explain "SWBD".

---

## Referee Comment (RC3) · Anonymous Referee #3 · 12 Mar 2018

This paper presented a comprehensive and highly valuable satellite water quality dataset derived from MERIS sensor for 300 lakes around the world. The dataset is a great addition to collection of publicly available datasets for the international community of inland water quality research and applications. The manuscript is well written. I have several minor points for the authors to consider.

1. It is concluded that "the Diversity II dataset is the first ... information source for inland water quality ..." (p. 10, line 27). I would be more cautious to make such a statement. It is somewhat subjective to define the standards based on which something can be called the first. Someone else may argue that datasets of the same kind have been around for many years. So I suggest to reword this statement.

2. Likewise, I would also not claim that the dataset "represents the state-of-the-art"

[Figure]

(p. 11, line 4), which again is subjective and invites arguments. When it comes to water quality algorithms, I do not think that the scientific community have reached an agreement on which ones are the state of the art yet. Instead, all algorithms have their "plus and cons". Therefore, I would remove this statement. In addition, I would add discussions in section 4.1 and 4.2 about key assumptions of each water quality algorithm and atmospheric correction algorithm used in developing this dataset, and implications about when the Diversity II products are most robust, and when they should be used with greater caution. This way the data users will have a clearer idea about the strength and limitations of the dataset and have more confidence in using the data.

3. I note that DINEOF was used to fill data gaps (section 3.3). Is there a place in the dataset to tag the DINEOF-interpolated data so that they can be differentiated from "real" data measured by satellite? If not I would strongly recommend to add such a funtionality, which will increase user's confidence in data quality.

---

## Editor Comment (EC1) · F. Huettmann (Editor) · 29 Mar 2018

Dear Authors, Colleagues,

thanks, I have usually not much to say beyond what the three reviewers commented. I await for a reply to those comments first.

But just to add to the existing feedback and to be addressed early on: I fully agree with the need for a confidence/error metric for such data and analysis. Secondly, I would probably call the physical metric an 'index' (e.g. interpreted, based on a computation and with relative units); having a quality flag is meaningful for data points.

The more those details get described and addressed probably the better it is for the users (and the authors eventually).

[Figure]

Thanks, more then with a reply by the authors.

Falk Huettmann, Associate Editor PhD, Associate Professor, University of Alaska Fairbanks

---

## Author Comment (AC1) · 17 Apr 2018

Dear referee 1,

Thank you very much for your interest in our manuscript and your valuable comments. We agree that data accuracy is of high concern, and we will revise the manuscript with a section that summarizes our validation efforts (using material that was released in Chapter 4.1.4 of Odermatt et al., 2015; http://www.diversity2.info/products/documents/DEL5/DIV2_Algorithm_Theoretical_Basis_Document_v2.4.pdf) Note however that we can only give meaningful accuracy estimates for the chlorophyll products.

Even in the case of chlorophyll, the validation is limited by: [1] The scarce availability of

reference data, [2] the inconsistency of reference data across different lakes, in terms of estimation methods and protocols, quality control etc.; in most cases, reference data don't come with any uncertainty characterisation, which limits a quantitative validation, [3] the comparability of vertically discrete point-measurements and pixel-size, vertically integrated satellite observations, [4] the variations in pigment absorption efficiency, which introduce considerable uncertainty in the calculation from pigment absorption as estimated by the satellite to the pigment concentration measured in situ, and thus reduce absolute accuracy.

As far as the other parameters are concerned, the main limitation is that we found even less reference data than for chlorophyll. This applies to our surprise also to TSM and turbidity, for which we found some, but methodologically very diverse and not enough reference data. The availability of CDOM reference measurements was even further from being globally representative. In the case of immersed_cyanobacteria, floating_cyanobacteria, floating_vegetation and owt_cc_dominant, we provide indicators that were defined based on remote sensing reflectance, without a suitable counterpart that is available from routine monitoring measurements. And finally for LSWT we redistribute monthly products that were produced by others. Of course their accuracy estimates can be cited, as well as accuracy estimates available from independent validation studies of the optical algorithms. All other limitations discussed here will be included in the corresponding section of the revised manuscript.

We are aware that much more useful reference data exists, but often these are stored in national databases and with often complicated accessibility. Its collection is laborious to the degree that even UNEP's GEMStat database struggles to achieve global representativeness, therefore this task is beyond our possibilities.

As far as your editorial comments are concerned, please take note that 'FR' is already introduced on page 2, row 14. The order of references by the same authors is done automatically by the Zotero citation style repository for ESSD. The term 'complementing' seems correct. All other short comments are acknowledged and will be accounted
for.

Best regards, The authors

Odermatt, D., Gangkofner, U., Ratzmann, G., Ruescas, A.B., Stelzer, K., Philipson, P., and Brockmann, C. (2015). Algorithm Theoretic Baseline Document v2.4 (ESA DUE Project Diversity II).
* * *

---

## Author Comment (AC2) · 17 Apr 2018

Dear referee 2,

Thank you very much for your interest in our manuscript and your valuable comments. One of your main concerns, data accuracy, is already referred to in our reply to referee 1, and we kindly ask you to check our suggestions on how to deal with this requirement. We agree with you that, for a specific lake or application, a detailed presentation of the respective data availability would be useful. However, this would require a detailed and long listing per lake if done at the level of detail you recommend. In our view this would be too complex, in particular because it differs for most of the water quality parameters due to different valid pixel expressions. But we suggest to include a global

map, where coloured point markers at lake centroid locations indicate the fraction of monthly products where observations are available for at least 50% of the lake for one of the products, probably turbidity_cc_mean or owt_cc_dominant_class.

Providing remote-sensing reflectance is in our opinion out of scope. Our processing chain works on temporal aggregates of derived parameters, like chlorophyll, with the smallest period of monthly averages. While temporal averaging of scalar quantities is correct, it is not appropriate for reflectances, where the average of two observed spectra could result in a spectrum that can never be observed. Complementing the dataset with a surface reflectance product is a valid request, however, it requires a dedicated algorithm, e.g. to select the most representative spectrum. Our research was aiming at an audience that is looking for off-the-shelf water quality information, not an audience that is willing to perform own retrievals and thus we did not develop such an algorithm.

As far as the technical corrections are concerned, we acknowledge very much the reviewer's visual acuity that revealed two faulty cross references. We suggest to restructure the sub sections Conclusions and Limitations as main sections (6. Conclusions, 7. Limitations). We think that the reference and describing sentence on SWBD are sufficient to describe the use in the given context, but are happy to consider specific explanations required by the reviewer.

Best regards, The authors

---

## Author Comment (AC3) · 17 Apr 2018

Author response:

Dear referee,

Thank you very much for your interest in our manuscript and your valuable comments. We are quite concerned not to make unjustified claims, and appreciate an external opinion on them.

1) The full statement is "The Diversity II dataset is the first globally representative, temporally resolved and methodologically consistent information source for inland water quality dynamics from satellite Earth observations." We chose this phrase because we pointed at the work by Sayers et al. (2015), who was obviously the first to publish a

global snapshot. We completed the upload of our products to Pangaea in February 2017, and submitted the present manuscript in January 2018. We have communicated our project progress within the scientific community since 2012. We announced the present discussion, among other channels, through the GEO Water Quality initiative (https://www.geoaquawatch.org/news-events/), which is the largest group of specialists in the field of water quality remote sensing. We never received a hint on a comparable dataset there or elsewhere. Therefore, we think our claim is well justified. Of course we would change it if you could point at a relevant source.

2) In our opinion, 'state-of-the-art' is a relatively wide term, which can simultaneously apply to several methods, in particular when methodological consolidation is as weak as in our field. Therefore, we don't consider the use of this term critical, but we could replace it with a different attribute, e.g. "widely used and validated by several independent users" if the reviewer's concern persists. Concerning the conceptual assumptions for each algorithm we agree that more detail will be of interest for the users and are happy to revise and expand chapter 6.2 accordingly.

3) The DINEOF-based LSWT products we included are adopted from the ARC-Lake database, where a variety of product versions are available, including the actual observations. We will insert a remark to point this option out to potential users, and therewith address your worthwhile request.

Best regards, The authors

---

## Author Comment (AC4) · 17 Apr 2018

Dear Professor Huettmann,

Thank you very much for your feedback and for organizing the review process.

Please consider the validation report mentioned in the reply to reviewer 1 and our suggestions for the content of a product accuracy chapter. Concerning the use of physical metrics, we think your suggestion could be a pragmatic approach to prevent false expectations in terms of absolute accuracy. However, all the retrieval methods we used were developed and published by other scientists, and the output values are by their definition concentrations. I think the situation is comparable to the use of fluorescence probes, which measure an optical quantity that is related but not equal to

pigment concentration. Nevertheless, the output data is in pigment concentration units, because the signal interpretation procedure is defined to retrieve them.

I'm looking forward to your reply, and I'm committed to faster responding in the future after managing a job change in the past weeks.

Best regards, Daniel